# *PhCHS5* and *PhF3′5′H* Genes Over-Expression in *Petunia* (*Petunia hybrida*) and *Phalaenopsis* (*Phalaenopsis aphrodite*) Regulate Flower Color and Branch Number

**DOI:** 10.3390/plants12112204

**Published:** 2023-06-02

**Authors:** Yuxia Lou, Qiyu Zhang, Qingyu Xu, Xinyu Yu, Wenxin Wang, Ruonan Gai, Feng Ming

**Affiliations:** 1Development Centre of Plant Germplasm Resources, College of Life Sciences, Shanghai Normal University, Shanghai 200234, China; 2Shanghai Key Laboratory of Plant Molecular Sciences, College of Life Sciences, Shanghai Normal University, Shanghai 200234, China

**Keywords:** *Phalaenopsis*, anthocyanin, chalcone synthase, Flavonoid 3′,5′-hydroxylase, *Petunia*

## Abstract

Flower breeders are continually refining their methods for producing high-quality flowers. *Phalaenopsis* species are considered the most important commercially grown orchids. Advances in genetic engineering technology have provided researchers with new tools that can be used along with traditional breeding methods to enhance floral traits and quality. However, the application of molecular techniques for the breeding of new *Phalaenopsis* species has been relatively rare. In this study, we constructed recombinant plasmids carrying flower color-related genes, *Phalaenopsis* Chalcone synthase (*PhCHS5*) and/or Flavonoid 3′,5′-hydroxylase (*PhF3′5′H*). These genes were transformed into both *Petunia* and *Phalaenopsis* plants using a gene gun or an *Agrobacterium tumefaciens*-based method. Compared with WT, *35S::PhCHS5* and *35S::PhF3′5′H* both had deeper color and higher anthocyanin content in *Petunia* plants. Additionally, a phenotypic comparison with wild-type controls indicated the *PhCHS5* or *PhF3′5′H*-transgenic *Phalaenopsis* produced more branches, petals, and labial petals. Moreover, *PhCHS5* or *PhF3′5′H*-transgenic *Phalaenopsis* both showed deepened lip color, compared with the control. However, the intensity of the coloration of the *Phalaenopsis* lips decreased when protocorms were co-transformed with both *PhCHS5* and *PhF3′5′H.* The results of this study confirm that *PhCHS5* and *PhF3′5′H* affect flower color in *Phalaenopsis* and may be relevant for the breeding of new orchid varieties with desirable flowering traits.

## 1. Introduction

Orchidaceae is the largest angiosperm family, with more than 29,000 species, accounting for 7–10% of all angiosperms. A previous study confirmed that the orchids form one of the largest and most evolved and diverse taxa in the plant kingdom [1]. Floriculture is one of the most economically important industries, with *Phalaenopsis hybrida* currently the most desired ornamental orchids worldwide. High-quality flowers can enhance the ornamental value of plants, while also promoting the economic growth of the floral industry. Over the past decade, researchers have explored the applicability of genetic engineering for modifying floricultural plants [2,3,4]. Flower color is one of the most important quality traits among angiosperms, and most insect-pollinated plants have large, brightly colored petals, which influence the pollination behavior of insects. Additionally, the anthocyanin content of plants is related to the success of sexual reproduction and the continued propagation of species [5]. Approximately 33% of Orchidaceae species rely on deceptive pollination, with their petal colors and patterns important for attracting insects [6]. Floral fragrance in the orchid family also attracts insect pollination [7]. *Phalaenopsis* species are excellent model plants for studying the molecular mechanisms regulating anthocyanin biosynthesis because of their natural variations in flower color. The availability of the complete *Phalaenopsis equestris* genome sequence and some orchid transcriptase gene sequencing data has facilitated the identification of candidate genes [8].

In this study, we investigated the key enzymes in the anthocyanin biosynthesis pathway. Anthocyanin biosynthesis is regulated by various enzymes, including chalcone synthase (CHS) (EC.2.3.1.74). The production of CHS determines the flavonoid diversity and content in plants, with implications for flower colors and/or patterns [9]. Thus, silencing, overexpressing, or mutating *CHS* genes will directly or indirectly affect anthocyanin biosynthesis and floral coloration [10]. In an earlier study, the *CHS* gene of *Torenia hybrida* was silenced by RNA interference (RNAi), which caused the flowers to change from blue to white or pale colors [11]. Chalcone synthase, which is the first enzyme in the anthocyanin biosynthesis pathway, catalyzes the synthesis of chalcone, which is converted to dihydrokaempferol (DHK) by consecutive reactions catalyzed by chalcone isomerase (CHI) and flavanone 3-hydroxylase (F3H). Flavonoid 3′-hydroxylase (F3′H) and flavonoid 3′,5′-hydroxylase (F3′5′H) respectively convert DHK to dihydroquercetin and dihydromyricetin, which are two types of dihydroflavonols [12]. Therefore, the F3H, F3′H, and F3′5′H activities determine anthocyanin structures, making them important enzymes for the coloration of flowers [13].

Previous research revealed F3′5′H belongs to the CYP75B subfamily of the cytochrome P450 superfamily, and is specifically produced in flowers, wherein it is crucial for the blue coloration of petals [14]. The regulation of enzymes and transcription factors involved in anthocyanin biosynthesis can significantly alter flower colors. The insertion of the *F3′5′H* gene into various flowers, such as roses and carnations, results in delphinidin production, which turns flowers purple or violet [15]. In chrysanthemums transformed with the butterfly pea gene *CtA3′5′GT* and the Canterbury bells gene *CamF3′5′H*, which contribute to anthocyanin structural modifications via B-ring hydroxylations and glucosylations, delphinidin-based anthocyanin acumulate to produce blue flowers [16]. Engineering of the rose flavonoid biosynthetic pathway successfully generated blue-hued flowers accumulating delphinidin [13]. A gene expression analysis by quantitative real-time PCR indicated that *F3′5′H* gene expression is followed by anthocyanin accumulation. Additionally, F3′5′H activity is positively correlated with the anthocyanin content [17,18]. isolated the cytochrome P450 family *F3′5′H* gene involved in the synthesis of anthocyanin in blue and fuchsia *Phalaenopsis* flowers, after which they transferred the gene into a red-flowered *Phalaenopsis* species. An analysis of the genetically modified flowers revealed that the petals changed from red to purple in 48 h, indicating that the *F3′5′H* gene can affect anthocyanin biosynthesis [18]. Additionally, blue-hued carnations or roses can be produced by eliminating the competition from the endogenous enzymes F3′H, DFR, and FLS. Accordingly, down-regulating the expression of the genes encoding these enzymes and inserting the *F3′5′H* gene into the genome can result in the accumulation of substantial amounts of delphinidin-type anthocyanin [19]. For example, the genetic engineering of chrysanthemum via the RNAi-based silencing of the endogenous *F3′H* gene and the insertion of the pansy *F3′5′H* gene coding region under the control of the rose flower-specific *CHS* promoter reportedly results in transgenic plants that produce blue flowers [20]. Additionally, RNAi technology was applied to target the gentian *F3′5′H* and *5/3′AT* genes, which changed the flower color from blue to light blue [21].

We previously cloned three *Phalaenopsis* CHS-encoding genes (*PhCHS3*, *PhCHS4*, and *PhCHS5*), and confirmed that the *PhCHS5* expression level is related to the anthocyanidin accumulation of the petals and lip during the floral pigment accumulation period [22]. Additionally, we cloned the *PhF3′5′H* gene, which is transcribed during a late petal development stage, coinciding with the anthocyanin production in petals [23].

*Petunia hybrida* is an important horticultural ornamental plant species, which has long been used as a genetic model system because its rapid growth and obvious biological characteristics make it easy to conduct molecular analyses. Moreover, it is well suited for comparative genomics studies. Thus, *P. hybrida* has become a useful model plant species for investigating flowers [24].

In the current study, we transferred the flower color-related *Phalaenopsis* genes (*PhCHS5* and *PhF3′5′H*) to *Petunia*, after which the phenotypes and anthocyanin contents of the transgenic plants were analyzed. Additionally, transforming *Phalaenopsis* with endogenous *PhF3′5′H* or *PhCHS5* showed deepened lip color. However, a co-transformation with both genes resulted in a faded lip color. Moreover, transgenic *Phalaenopsis* plants transformed with *PhCHS5*, *PhF3′5′H*, or *PhCHS5 + PhF3′5′H* exhibited morphological changes, including increased leaf differentiation and the production of multiple heads and many branches. The characterization of the *PhCHS5* and *PhF3′5′H* genes described herein may be important for improving the flower color quality of new varieties of *Phalaenopsis* and other ornamental plants.

## 2. Results

### 2.1. Phylogenetic Analysis of CHS5 and F3′5′H in Phalaenopsis Species

Anthocyanin are the colored end products of flavonoid synthesis. Flavonoid-3′5′-hydroxylase is the key enzyme in the synthesis of 3′5′-hydroxyanthocyanin; it catalyzes the hydroxylation of flavonoid at 3′, 5′ position of B ring to produce blue delphinide; chalcone synthase (CHS) is the first specific enzyme in the flavonoid synthesis pathway, which is an important rate limiting step in the whole flavonoid synthesis pathway. For further study and analysis of *PhF3′5′H* and *PhCHS5*, we selected 15 different species for cluster analysis. The result showed that F3′5′Hs from different species are mainly divided into two groups, monocotyledons and dicotyledons (Appendix A). The phylogenetic relationship between *PhF3′5′H* and Dendrobium moniliforme is the closest, and the nucleotide similarity rate is up to 82%. *CHS5* in *Phalaenopsis* clustered together with *Oryza sativa*, *Zea mays*, *Dendrobium hybrid* cultivar, *Cymbidium hybrid* cultivar, and *Bromheadia finlaysoniana*. Among them, *PhCHS5* had the closest genetic relationship with that in *Bromheadia finlaysoniana*, and the nucleotide similarity rate reached 95% (Appendix A).

The *PhCHS5* open reading frame (ORF) with an *Xba* I restriction site at the 5′ and 3′ ends was generated by PCR as previously described. The amplified 1404-bp *PhCHS5* fragment was inserted into the pCAMBIA2301 vector (Figure 1a). The resulting recombinant plasmid, pCAMBIA2301-sense-*CHS5*, was inserted into *A. tumefaciens* strain GV3101 cells. Additionally, the *PhF3′5′H* ORF with the *Xho* I and *Bgl* II restriction sites added to the 5′ and 3′ ends, respectively, was also amplified by PCR. *A. tumefaciens* cells were transformed with the assembled plasmids carrying the target fragment. The transformation was confirmed by the amplification of specific fragments (approximately 1.7 kb) by PCR (Figure 1b). The *PhCHS5* sequence was inserted into *Petunia* plants via an *A. tumefaciens*-mediated transformation method (Figure 1c,d). The *PhF3′5′H*-transformed *Petunia* plants were differentiated and rooted (Figure 1e,f). A PCR was completed with the *NPTII* gene primers (Figure 1g), and the amplified fragment was the expected size (approximately 600 bp), implying the transformation was successful. Similarly, a PCR analysis of the transgenic *Petunia* with the *F3′5′H* primers revealed an amplified fragment that was consistent with the target fragment, indicating the *Petunia* plants were correctly transformed (Figure 1h).

### 2.2. Analyses of the Phenotypes and Anthocyanin Contents of Genetically Modified Petunia hybrida

In order to further study how *PhCHS5* and *PhF3′5′H* regulate flower color formation, *Petunia hybrida* was overexpressed with *PhCHS5* and *PhF3′5′H* to obtain genetically-modified lines. Compared with WT, *PhCHS5*-overexpressing lines (*PhCHS5-1* and *PhCHS5-2*) had deeper flower color and higher anthocyanin content (Figure 2a). Two genetically-modified lines *35S::PhF3′5′H-1* and *35S::PhF3′5′H-2* were obtained by overexpression of *PhF3′5′H*. Compared with WT, *35S::PhF3′5′H-1* and *35S::PhF3′5′H-2* both had deeper color and higher anthocyanin content (Figure 2b,c). The above results indicated that *PhCHS5* and *PhF3′5′H* could deepen flower color and increase anthocyanin content in the pathway of flower color formation.

### 2.3. Induction and Cultivation of Phalaenopsis Protocorms

The *Phalaenopsis* protocorms required for the transgenic study were mainly generated from axillary bud explants. Specifically, the *Phalaenopsis* axillary bud explants were added to the *Phalaenopsis* induction medium in culture bottles. To produce high-quality materials and minimize contamination, four or five *Phalaenopsis* axillary bud segments were added to each culture bottle. Within two weeks of inoculation, the axillary buds began to swell (Figure 3a). The above materials were subcultured twice under aseptic conditions, after which the axillary buds began to elongate and produce leaves (Figure 3b). The leaves that grew to approximately 1 cm long were cut and transferred to the same induction medium. Many protocorms formed after two months (Figure 3c). Additionally, we examined the effects of different BA and NAA concentrations on the induction of protocorms. Ensuring the auxin and cytokinin contents are balanced is critical for plant morphological development. The protocorm induction efficiency of YD3 (3 ppm BA and 0.1 ppm NAA) was 80% (Table 1). In the YD3 medium, the original stem formed buds within 4 weeks (Figure 3d,e), which was efficient for the production of genetically modified materials. Increasing the NAA concentration (Figure 3f,g) or decreasing the BA concentration (Figure 3h,i) inhibited the production of protocorms and buds formation.

### 2.4. Screening of PhCHS5 and PhF3′5′H in Transformed Phalaenopsis Protocorms

The *PhCHS5* and *PhF3′5′H* genes were cloned and determined to be involved in anthocyanin biosynthesis. Thus, we analyzed the expression of these two genes to clarify their roles related to flower coloration. The pCAMBIA2301-*PhCHS5* and pCAMBIA2301-*PhF3′5′H* recombinant plasmids were constructed for the subsequent analysis of the overexpression of the genes in the *Phalaenopsis* hybrid ‘Formosa Rose’ background through *A. tumefaciens* infection and gene gun. The *PhCHS5*-transformed protocorms and *PhF3′5′H*-transformed protocorms were screened for cefotaxim resistance. GUS staining showed *PhCHS5* and *PhF3′5′H* were successfully transferred into *Phalaenopsis* protocorms (Figure 4a,c). Those that grew well in the presence of cefotaxim were cultured to eventually produce leaves (Appendix A). A PCR analysis of the leaf in transgenic *Phalaenopsis* with the *Kan* gene primers revealed an amplified fragment that was consistent with the target fragment, indicating *Phalaenopsis* plants were correctly transformed (Figure 4b,d).

### 2.5. Regulatory Effects of PhCHS5 and PhF3′5′H Expression on the Branching of Phalaenopsis Stems and the Color of Phalaenopsis Lips

Transgenic seedlings with *PhCHS5*, *PhF3′5′H*, or *PhCHS5* + *PhF3′5′H* grew well after the rooting, transplanting, and cultivation of whole plants (Appendix A). After two months’ growth, partially of the *PhCHS5*-transformed plants (Appendix A), a few of the *PhF3′5′H*-transformed plants (Appendix A) and some of *PhCHS5* + *PhF3′5′H* transgenic plants were observed to grow abnormally, with asymmetrical plants and diverse leaf shapes. However, most of the transformed plants with normal shapes and grew well for further evaluation.

The splitting of the single stem of *PhCHS5*-transgenic plants resulted in multiple branches and *Phalaenopsis* plants with multiple heads. The stem of the *PhF3′5′H*-transformed plants also appeared to split into two (Figure 5a,b), with an increasing number of flowers with increases in the number of flowering branches (Figure 5a,b). The transgenic plants carrying *PhCHS5* and *PhF3′5′H* produced many branches and multiple heads (Figure 5a,b). Additionally, *PhCHS5* or *PhF3′5′H*-transgenic plants both showed deepened lip color, compared with the control (Figure 5c). However, the intensity of the coloration of the *Phalaenopsis* lips decreased when protocorms were co-transformed with both *PhCHS5* and *PhF3′5′H* (Figure 5c). Accordingly, both genes are likely necessary for the regulation of *Phalaenopsis* lip coloration and branch development.

## 3. Materials and Methods

### 3.1. Materials and Growth Conditions

*Petunia* is laboratory reserved variety with red petals which were grown in a greenhouse of Shanghai Normal University (SHNU) and *Phalaenopsis* hybrid cv. Formosa roses with white petals are collected from the Vegetable and Horticulture Research Institute of the Shanghai Academy of Agricultural Sciences. *Petunia* was cultivated at a temperature of 24–26 °C with light for 16 h and dark for 8 h, the relative humidity is maintained at 74%, and the light intensity is 5000 lux. *Phalaenopsis* was cultured at a temperature of 18–20 °C with light for 16 h and dark for 8 h with a light intensity of 2500 lux and a relative humidity maintained at 80%.

### 3.2. Preliminary Treatment and Cultivation of Phalaenopsis Tissue Culture

The well-grown shoot segments of *Phalaenopsis* hybrid (Formosa rosa) were selected as materials, and the leaves outside the axillary buds were carefully peeled off and cut into 1 cm stem segments. Soak in soapy water and wash it many times. Rinse in running tap water and dry. It was immersed and sterilized by 5% NaClO on a clean bench for 15 min, and then rinsed 6 times with sterile water. The materials are cultured in the culture room after being inoculated into the sterilized medium. Medium formulation of induction and differentiation were shown in Appendix A.

### 3.3. Construction of PhCHS5 and PhF3′5′H Prokaryotic Expression Vectors

PCR was performed using forward primers for the 5′end of *PhCHS5* is 5′GCTCTAGAGGAGAGGGAGTTAATGGC3′ and the 3′end reverse primer is 5′GCATTTTGTGGTTTTATTGGACT3′ (*Xba* I linker is added to both ends of the forward primer and reverse primer). The forward primer for the 5′end of *PhF3′5′H* is 5′CCGCTCGAGATGTCCATCTTCCTCATCATCACACACACCC3′, and the reverse primer at the 3′end of *PhF3′5′H* is 5′GAAGATCTTCAAACAACCCCATACGCCGCCG3′ (with *Bgl* II restriction site) (Shanghai Bioengineering Co., Ltd., located in Shanghai, China). Afterwards, the PCR products of *PhCHS5* and the pCAMBIA2301 vector were double-digested with *Xba* I, *Xho* I; the PCR products of *PhF3′5′H* and the pCAMBIA2301 vector were double-digested with *Xho* I, *Bgl* II.

### 3.4. Agrobacterium-Mediated Leaf Disc Method Transgenic to Obtain Transgenic Petunia Expressing PhCHS5 and PhF3′5′H Gene

By transgenic leaf discs, *Petunia* leaves were cut into 0.8 × 0.8 cm (1 cm) squares, placed in MS0 solution containing Agrobacterium for infection, and incubated in the dark on MS1 at 22 °C for three days [25]. Afterward, the bacterial cells are washed and excess water is absorbed with absorbent paper, the leaves are transferred to MS2 and cultured at 25 °C for subculture every 20 days, and the Seedlings differentiated from calluses are thoroughly washed with sterile distilled water after they grow, and then transferred to plastic pots containing peat-based soil. Seedlings are incubated in a growth chamber at 25 ± 2 °C at 85% relative humidity for 2–3 days. The seedlings are then transferred to plastic pots filled with peat-based soil and grown in greenhouses under controlled conditions. The medium formula is as follows:

Differentiation medium (MS1) formulation: MS, 6-BA 1 mg/L, NAA 0.1 mg/L, plant gel concentration 0.25%, sucrose final concentration 3%, pH 6.0. Differentiation liquid medium (MS0) formulation is without plant gel concentration. Rooting medium formulation: 1/2MS, plant gel concentration 0.25%, sucrose final concentration 3%, pH 5.8. Resistance screening medium formulation: on the basis of normal differentiation and rooting medium for → mula, Kan 25 mg/mL + Cef 250 mg/mL (MS2) or Kan 25 mg/mL + Cef 500 mg/mL (MS3) were added respectively.

### 3.5. Construction of Transgenic Phalaenopsis by Gene Gun

After osmotic treatment for 1 h, the protocorms of *Phalaenopsis* were cut into pieces and pre-cultured in a 60 mm Petri dish containing YD3 medium (see Appendix A). The plasmid was purified by QIAGEN Plasmid Purification Kit (Qiagen, Shanghai, China) and precipitated on gold particles 1.0-(m). In a vacuum environment, the Bio-Rad system was used, and bombarded with 1100 kPa helium, bombarded twice per dish. The protocoms after bombardment were transferred to new YD3, and cultures (containing corresponding antibiotics) were screened after 2–10 days. Data were obtained from at least three biological replicates and blank controls per sample.

### 3.6. Construction of Transgenic Phalaenopsis by Agrobacterium Infection

Shake overnight (not less than 7 h, 28 °C, 200 rpm). Make several holes in the protocom used for transgene. The protocol was placed in the bacterial solution, and the bacteria were shaken for 30 min. Appropriately remove excess bacteria. The protocom was clamped to YD3 (see Appendix A) solid medium covered with a layer of sterile filter paper and cultured for 3 days in the dark. After three days, the excess bacterial solution was washed away, and the protocom was washed by 3–5 times by YD3T with antibiotics, at last, were placed on a medium of YD3 (see Appendix A) + cef (250 mg/L) for 6 weeks. The newly grown protocorm-likebodies (PLB) were then transferred to YD3 and YDR rooting medium (supplemented with antibiotics) for Proliferation and rooting.

### 3.7. Analysis of Anthocyanin in the Petals of Transgenic Petunia

Put the petals in liquid nitrogen and grind thoroughly. Add 1 mL of the prepared acidic methanol (methanol: HCl = 24:1) to each test tube and let it shake well on the shaker. To disperse petal fragments. At 4 °C and 100 rpm, the vibrator vibrates for 2 days, during which it vibrates twice on the vibrator. After that, the anthocyanin content was measured, and the atmosphere was diluted to 2 mL with 800 mL, and then the absorbance of A530 and A657 was measured with a spectrophotometer to calculate the anthocyanin content (U/g).

### 3.8. Molecular Identification and Phenotype Analysis of Transgenic Phalaenopsis Strains

*Kan* gene was used in PCR to screen transgenic strains, designing primers based on its genetic sequences, The forward primer for the 5′end of *Kan* is 5′ATTTTCTCCCAATCAGGCTTGATCC3′, and the reverse primer at the 3′end of *Kan* is 5′CACCTATGATGTGGAACGGGAAAAG3′. DNA extraction was performed by the method [26]. After the plant grows to a certain stage, it is subjected to flower treatment to perform plant phenotype identification.

### 3.9. Statistical Analysis

The date including anthocyanin content, rate of protocorm and branch number was used in *t*-test analysis and the Dumcans multiplerenge test method.

## 4. Discussion

Both *CHS5* and *F3′5′H* are key enzymes contributing to anthocyanin biosynthesis in plants, with crucial regulatory functions related to plant coloration. To date, the effects of *CHS* and *F3′5′H* genes on plant colors have mostly been investigated in *Rosa rugosa* Thunb and *Petunia hybrida* Vilm [27]. Due to their varied colors, orchids have become the ideal plant materials for studying the mechanism underlying the development of flower colors. Increases in the commercial production of *Phalaenopsis* species have coincided with increasing molecular research on these important orchids. However, these species have not been as well characterized as some other widely grown crops, including rice and corn. In China, much of the relevant research currently involves *Phalaenopsis* tissue cultures, but there are few reports describing research on the genetic basis of flower coloration [28].

The *F3′5′H* gene has been studied by several researchers. For example, the overexpression of the violet *F3′5′H* gene in *Rosa chinensis* ‘Lavande’ increased the delphinidin accumulation by up to 44.2% [29]. Nakano and Tanaka produced purple lilies by generating plants constitutively expressing the *Campanula F3′5′H* gene under the control of the CaMV 35S promoter [19,30]. In another study, the expression of the *Phalaenopsis F3′5′H* gene in lily flowers caused the petals to change from pink to blue [31]. Additionally, purple carnations and roses accumulating delphinidin-based anthocyanin following the introduction of the *F3′5′H* gene have been commercialized globally [32]. The results of the current study are similar to those of earlier investigations. The flower colors of the *PhF3′5′H*-transformed *Petunia* were relatively intense (Figure 2), implying that *PhF3′5′H* likely induces the accumulation of anthocyanin.

In an earlier investigation, the expression of antisense *CHS* genes in *Petunia* led to the production of completely white flowers [33]. Additionally, the overexpression of the *CHS* gene was expected to deepen the flower colors. In the current study, *PhCHS5*-transformed *Petunia* showed deeper flower color, while *Phalaenopsis* plants exhibited deepened lip color (Figure 5c), which was consistent with the expectation. To our interest, the intensity of the coloration of the *Phalaenopsis* lips decreased when protocorms were co-transformed with both *PhCHS5* and *PhF3′5′H*, compared with the plants transformed with *PhCHS5* or *PhF3′5′H* alone (Figure 5c), which suggested that there might be conjugate effect between *PhCHS5* and *PhF3′5′H*, in which interaction cofactor between them should be further studied.

The transgenes analyzed in this study had no effect on the production of white petals (Figure 5), possibly because of an endogenous mechanism that prevents anthocyanin synthesis and accumulation. In a previous study, a rare mutation in white flowers due to *DFR* inactivation was detected in a *Solanaceae* species [34]. Most *Caryophyllales* species are unable to accumulate anthocyanin because of a lack of DFR and ANS activity [35]. *Phalaenopsis* ‘Formosa Rose’ petals may lack the substrate for anthocyanin biosynthesis or key enzymes (i.e., DFR and ANS), which may explain their inability to synthesize anthocyanin or their production of only colorless anthocyanin. However, our analysis of the flowers of *Phalaenopsis* plants transformed with both *PhCHS5* and *PhF3′5′H* revealed a deepening of the lip color, indicative of the accumulation of colored anthocyanin in the lip (Figure 5). Moreover, the transformation of the endogenous genes contributed to the increased anthocyanin accumulation. Accordingly, the expression levels of *PhCHS5* and *PhF3′5′H* may be related, possibly due to the regulatory effects of specific transcription factors.

In the current study, the application of exogenous and endogenous genes to synthesize anthocyanin was insufficient for enhancing the petal coloration of white *Phalaenopsis* flowers. There are published reports indicating that the transformation of *Phalaenopsis* plants producing pink flowers with the *Commelina communis F3′5′H* gene generates transgenic plants with bluish flowers [15,30]. Thus, we will need to evaluate the effects of pH, metal ions, transcription factors, and genetic modifications to determine how to obtain blue *Phalaenopsis* flowers. In *Petunia*, seven genetic loci (*PH1-PH7*) regulating the pH of petals have been identified [36]. Under acidic and alkaline conditions, the petals will be reddish and bluish, respectively. Metal ion transporters can also influence flower coloration. The tulip *TgVit1* gene and its homolog in *Centaurea cyanus* (*CcVit*) encode vacuolar iron transporters, which promote the accumulation of iron ions in the vacuole, resulting in blue petals [37]. Changes to the transcription factors associated with the petal anthocyanin biosynthesis pathway may alter flower colors. The results of these earlier investigations may be relevant for developing novel methods for improving flower colors. However, enhancing the floral coloration of *Phalaenopsis* species requires a more thorough elucidation of the underlying mechanism. The data derived from recent whole-genome sequencing efforts may provide a solid foundation for future comprehensive analyses of the mechanism mediating the formation of floral patterns. Additionally, like other non-model plant species, many ornamental plants are associated with specific problems adversely affecting molecular research, including a low genetic transformation efficiency, a long juvenile stage, the development of abnormal petals, and substantial variability in transgene expression. These problems must be overcome to facilitate the generation of new commercially valuable flower varieties. Our study findings described herein may provide the basis for future research on *Phalaenopsis* floral coloration, specifically regarding the effects of *PhF3′5′H* and *PhCHS5* on lip color.

## Figures and Tables

**Figure 1 plants-12-02204-f001:**
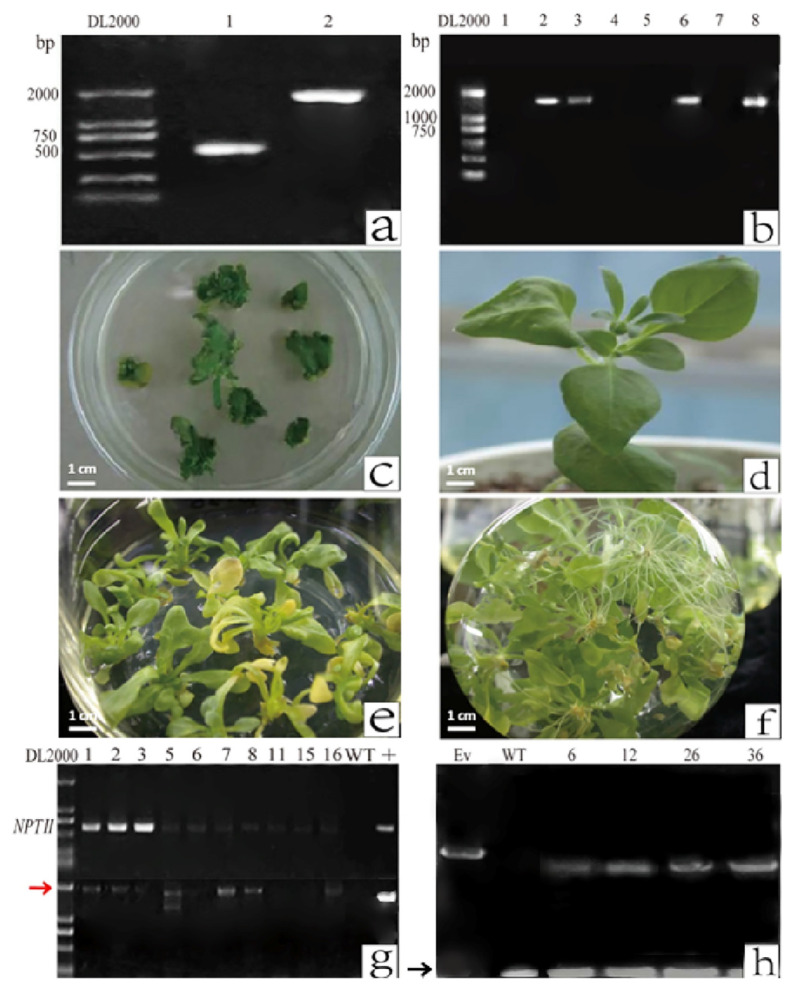
Construction of recombinant plasmids carrying the *PhCHS5* and *PhF3′5′H* sequences for the transformation of *Petunia*. (**a**) Confirmation of the construction of the pCAMBIA2301-sense-*CHS5* plasmid by PCR. Regarding the plasmid with *PhCHS5* in the forward direction, lane 1 presents the fragment amplified with the Cav35s-primerF and CHS-XR’ primers. Lane 2 presents the approximately 750-bp non-specific band amplified by PCR with the Cav35s-primerF and CHS-XF primers. (**b**) Confirmation of the transformation of *A. tumefaciens* cells with *PhF3′5′H* by PCR. Lanes 1 to 8 represent transformed *A. tumefaciens* colonies that were analyzed by PCR. (**c**) *Petunia* leaf explants during the transformation procedure. (**d**) Mature transgenic plant. (**e**) Transgenic *Petunia* tissue culture. (**f**) Rooting of transgenic *Petunia* plants. (**g**) Preliminary identification of *PhCHS5*-transformed *Petunia* plants by PCR. Each lane represents a different *Petunia* strain. The red arrow indicates the *PhCHS5* stripe; WT for wild type; +stand for vector with NPTII gene (**h**) Preliminary identification of *PhF3′5′H*-transformed *Petunia* plants by PCR. Each lane represents a different *Petunia* strain. The black arrow indicates the *PhF3′5′H* stripe; WT for wild type; Ev stand for vector with only NPTII gene without *PhF3′5′H*.

**Figure 2 plants-12-02204-f002:**
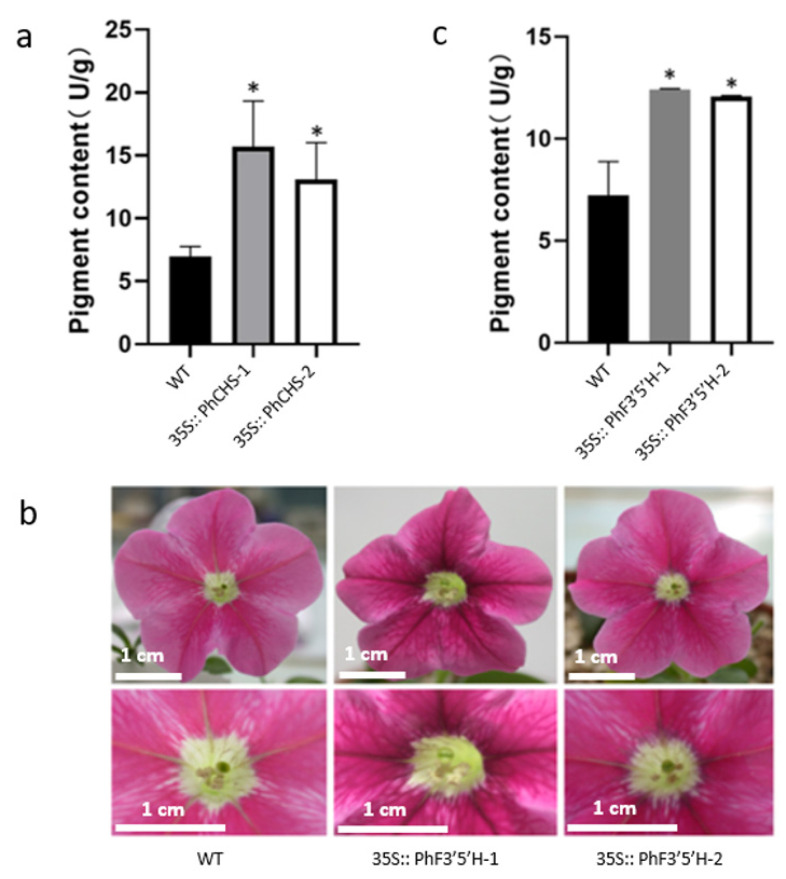
Phenotypic analysis and anthocyanin content determination of *Petunia* mutants *PhCHS5* and *PhF3′5′H*. (**a**) Anthocyanin content of *Petunia hybrida* transformed by *PhCHS5*. (**b**) The phenotype of *Petunia* transformed by *PhF3′5′H.* (**c**) Anthocyanin content of *Petunia hybrida* transformed by *PhF3′5′H*. WT: Wild Type. The data were expressed as mean ± standard error and repeated three times for each sample. * *p* < 0.05.

**Figure 3 plants-12-02204-f003:**
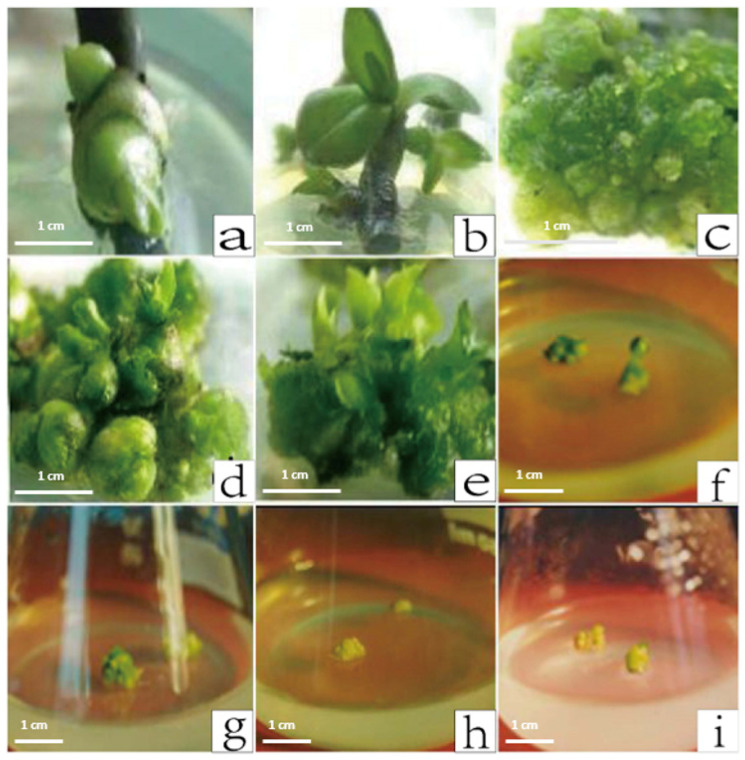
Induction of the *Phalaenopsis* protocorm. (**a**) Swollen axillary bud. (**b**) Seedling generated directly from an axillary bud. (**c**) Protocorm development. (**d**) Protocorm enlargement. (**e**) Seedling cluster. (**f**–**i**) Protocorms from YD4 (**f**,**g**), YD2 (**h**) and YD1 (**i**), respectively.

**Figure 4 plants-12-02204-f004:**
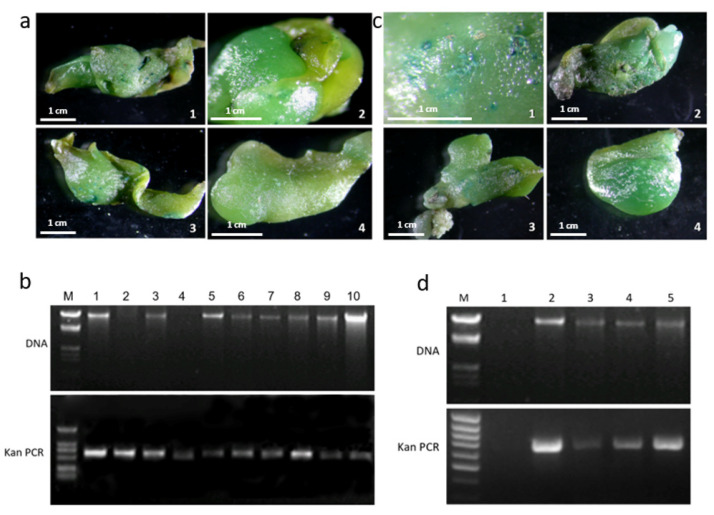
Screening of *Phalaenopsis* protocorms carrying *PhCHS5* and *PhF3′5′H* after transformations. (**a**) *PhCHS5*-transgenic *Phalaenopsis* protocorms. The image on the right is a close-focus image of the image on the left (**b**) Confirmation of the transformation of *Phalaenopsis* protocorms with *PhCHS5* by PCR for *Kan* gene.M stands for marker, 1–10 stands for transformation seedlings. (**c**) *PhF3′5′H*-transgenic *Phalaenopsis* protocorms. Figure 1 is a close-focus image, and Figure 2, Figure 3 and Figure 4 are three representative repeating telefocal images (**d**) Molecular confirmation of the presence of *PhF3′5′H* in the transformed samples by PCR for *Kan* gene. M stands for marker, 1–5 stands for transformation seedlings.

**Figure 5 plants-12-02204-f005:**
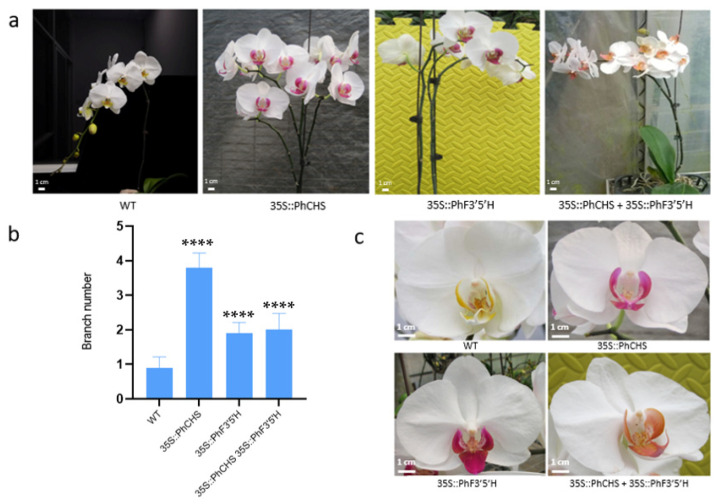
Phenotypes of the transgenic *Phalaenopsis* samples. (**a**,**b**) Branch number of the transgenic plants. (**c**) Flower phenotypes of the transgenic plants. WT: Wild Type. The data were expressed as mean ± standard error and repeated three times for each sample. **** *p* < 0.0001.

**Table 1 plants-12-02204-t001:** Effect of different hormones on the induction rate of protocorm.

Numbering	BA/NAA (ppm) Concentration	Protocorm Induction Rate (%)	Unit Axillary Buds Get the Number of Bushes
YD1	2/0.2	50 ± 1.2 ^b^	14 ± 1.4 ^b^
YD2	2/2.0	40 ± 2.6 ^c^	25 ± 3.4 ^a^
YD3	3/0.1	80 ± 2.0 ^a^	10 ± 1.6 ^bc^
YD4	3/2.0	10 ± 1.8 ^d^	4 ± 2.5 ^c^

All data is the mean of four repetitions. Using the Dumcans multiplerenge test method, different letters in the same column indicate significant differences (*p* < 0.05, *n* = 3).

## Data Availability

Not applicable.

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
