# Peer review of "PhCHS5 and PhF3′5′H Genes Over-Expression in Petunia (Petunia hybrida) and Phalaenopsis (Phalaenopsis aphrodite) Regulate Flower Color and Branch Number"

_plants, 2023, doi:10.3390/plants12112204_

Round 1

Reviewer 1 Report

In this manuscript, Phalaenopsis genes related to flower color (PhCHS5 and PhF3'5'H) were transformed into petunias, and the phenotype and anthocyanin content of the resulting transgenic plants were analyzed.

In addition, overexpression of PhF3'5'H or PhCHS5 genes in Phalaenopsis resulted in darker flower color.

However, the flower color of the transformants that overexpressed both genes at the same time faded.

In addition, transgenic Phalaenopsis plants transformed with PhCHS5, PhF3'5'H or PhCHS5 + PhF3'5'H showed morphological changes. Therefore, this manuscript is judged to be suitable for Plants journal.

This reviewer would like to point out a few things:

1. Fig. 1 would like to be moved to supplementary.

2. In the case of Fig 2, the entire electrophoresis image needs to be replaced.

3. In Fig 3A, there is no difference in flower color in the photo. So either replace it or remove it.

4. In Table 1, replace Protocorm induction rate (%) 2/2.0, 80 ± 2.0, 3/2.0

5. Replacing photos of B and D in Fig 5

6. In Discussion, consider in detail the functions and roles of the PhCHS5 and PhF3’5’H genes on the anthocyanin biosynthetic circuit.

Author Response

Dear editor and reviewers:

On behalf of my co-authors, we thank you very much for giving us an opportunity to revise our manuscript; we appreciate editors and reviewers very much for their positive and constructive comments and suggestions on our manuscript (plants-2304838). We have studied reviewer’s comments carefully and have made revision which marked in red in the paper. We have tried our best to revise our manuscript according to the comments. Attached please find the revised version, which we would like to submit for your kind consideration. We would like to express our great appreciation to you and reviewers for comments on our paper.

Looking forward to hearing from you.

Special thanks to you for your good comments!

Yours sincerely,

Feng Ming

The main corrections in the paper and the responds to the reviewer’s comments are as flowing:

Responds to the reviewer’s comments:

Reviewer #1:

  1. Fig. 1 would like to be moved to supplementary.

Response: Thanks for the suggestion. We have placed Figure 1 in supplementary Figure 1,and it should also be noted that there were six figures here, without Figures 7 and 8

  1. In the case of Fig 2, the entire electrophoresis image needs to be replaced.

Response: Thanks for the advice. We have replaced the electrophoresis plot in Figure 2 with a higher definition image.

  1. In Fig 3A, there is no difference in flower color in the photo. So either replace it or remove it.

Response: Thanks for the suggestion. We have deleted Figure 3A.

  1. In Table 1, replace Protocorm induction rate (%) 2/2.0, 80 ± 2.0, 3/2.0

Response: Thanks for the suggestion. We have replaced the Protocorm induction rate (%) in Table 1 with 2/2.0, 80 ± 2.0, 3/2.0.

  1. Replacing photos of B and D in Fig 5

Response: Thanks for kind suggestion. We have replaced the electrophoresis plot in Figure 5 with a higher definition picture.

  1. In Discussion, consider in detail the functions and roles of the PhCHS5 and PhF3’5’H genes on the anthocyanin biosynthetic circuit.

Response: Thanks a lot. It is a good suggestion. We add some evaluation in discussion part marked in red color.  

Reviewer 2 Report

Major:

1) I recommend that the authors should use some help of a native English speaker or send the Ms to an English Editing Service that proofreads scientific writing.

2) Authors should explain all used abbreviations, e.g. Line 21: “WT”.

3) Authors should improve used legends for figures, e.g. explain all used abbreviations, explain all investigated parameters, used statistical treatment

4) Authors should shorten the Introduction.

5) Line 165-173 is repetition of Line 149-156. Authors should avoid repetition.

6) Line Xba I, Xho I, and BglII, respectively. – for what is restrictase Xho I?

7) Line 218-221: include accession number of used actin gene.  

8) Author should improve all presented PCR data, e.g. include figure for actin PCT products. Also, I did not understand Fig. 2 g and h.

9 Authors should decrease the figure number, e.g. by combination Fig. 1 and 2; 5 and 6; 7 and 8.

10) Fig. 6b – include the statistical treatment.

Minor:

11) Fig. 2c,d,e,f; Fig. 3a,c; Fig. 4a-i; Fig. 5a,c; Fig. 6a,c: include bar 1 cm.

12) Line 144-147: - what is the illumination?

13) Line 269: “Agrobacterium tumefaciens”, correct to “A. tumefaciens”.

Author Response

Dear editor and reviewers:

On behalf of my co-authors, we thank you very much for giving us an opportunity to revise our manuscript; we appreciate editors and reviewers very much for their positive and constructive comments and suggestions on our manuscript (plants-2304838). We have studied reviewer’s comments carefully and have made revision which marked in red in the paper. We have tried our best to revise our manuscript according to the comments. Attached please find the revised version, which we would like to submit for your kind consideration. We would like to express our great appreciation to you and reviewers for comments on our paper.

Looking forward to hearing from you.

Special thanks to you for your good comments!

Yours sincerely,

Feng Ming

The main corrections in the paper and the responds to the reviewer’s comments are as flowing:

Responds to the reviewer’s comments:

Reviewer #2:

1) I recommend that the authors should use some help of a native English speaker or send the Ms to an English Editing Service that proofreads scientific writing.

Response: This MS was edited by Liwen Bianji (Edanz) (www.liwenbianji.cn/) for editing the English text of this manuscript. We added this sentence as following in Acknowledgement part: “We thank Liwen Bianji (Edanz) (www.liwenbianji.cn/) for editing the English text of a draft of this manuscript”.

2) Authors should explain all used abbreviations, e.g. Line 21: “WT”.

Response: Thanks for the suggestion. We have explained all the abbreviations in the legend.

3) Authors should improve used legends for figures, e.g. explain all used abbreviations, explain all investigated parameters, used statistical treatment

Response: Sorry for missing explanation for abbreviations and short of statistical treatment analysis for figure 5b. We checked the whole article and added explanation. There are no similar mistakes.

4) Authors should shorten the Introduction.

Response: Thanks for the suggestion. We have shortened the introduction to remove the introduction that deals with the regulatory role of MYB transcription factors on anthocyanin synthesis. we have re-written this part   according to your suggestion marked in red color in introduction part.

5) Line 165-173 is repetition of Line 149-156. Authors should avoid repetition.

Response: Thanks for the suggestion. You mentioned that the duplication represents two different tissue culture methods for plant material, and it needs to be preserved to avoid misunderstanding among readers.

6) Line Xba I, Xho I, and BglII, respectively. – for what is restrictase Xho I?

Response: Thanks for the suggestion. We're sorry that it is unclear here, so we correct it according to your suggestion. It was corrected as followings “Afterwards, the PCR products of PhCHS5 and the pCAMBIA2301 vector were double-digested with Xba I, Xho I; the PCR products of PhF3'5'H and the pCAMBIA2301 vector were double-digested with Xho I, BglII.”

7) Line 218-221: include accession number of used actin gene.  

Response: Thanks a lot! We are sorry that we made a mistake here.We correct it as following “Kan gene was used in PCR to screen transgenic strains, designing primers based on its genetic sequences, The forward primer for the 5'end of Kan is 5' ATTTTCTCCCAATCAGGCTTGATCC 3', and the reverse primer at the 3'end of Kan is 5’ CACCTATGATGTGGAACGGGAAAAG 3'. DNA extraction was performed by the method described in Chen F et al. (2019). After the plant grows to a certain stage, it is subjected to flower treatment to perform plant phenotype identification. ”

8) Author should improve all presented PCR data, e.g. include figure for actin PCT products. Also, I did not understand Fig. 2 g and h.

Response: Thanks for your suggestion. We have replaced the electrophoresis plot in Figure 2 and Figure 5 B and D with a higher definition image. Fig 2g and h represented PhCHS5-transformed and PhF3’5’H-transformed petunia plants were identified by Kan genes PCR detection.

9 ) Authors should decrease the figure number, e.g. by combination Fig. 1 and 2; 5 and 6; 7 and 8

Response: Thanks for the suggestion. We have placed Figure 1 in supplementary Figure 3. It should also be noted that there are only six figures in this article without figures 7 and 8

10) Fig. 6b – include the statistical treatment.

Response: Thanks a lot! We added statistical treatment analysis for Fig.6b now was marked in Fig.5b.

Minor:

11) Fig. 2c,d,e,f; Fig. 3a,c; Fig. 4a-i; Fig. 5a,c; Fig. 6a,c: include bar 1 cm.

Response: Thanks for the suggestion. We have already added the scale bars of these figures.

12) Line 144-147: - what is the illumination?

Response: Thanks for the suggestion. We are sorry that we have not clearly explained the lighting conditions for the cultivation of plant material, and we have added a description of the light intensity as following marked in red color:“Petunia was cultivated at a temperature of 24-26℃ with light for 16 hours and dark for 8 hours, the relative humidity is maintained at 74%, and the light intensity is 5000 lux. Phalaenopsis was cultured at a temperature of 18-20℃ with light for 16 h and dark for 8 h with a light intensity of 2500 lux and a relative humidity maintained at 80%”.

13) Line 269: “Agrobacterium tumefaciens”, correct to “A. tumefaciens”.

Response: Thanks for the suggestion. We have corrected "Agrobacterium tumemorcensis" to " A. tumefaciens " in the text.

Reviewer 3 Report

The manuscript describes construction of transgenic plants transformed with flavonoid bioisynthetic genes. This approach will contribute to adding breeding options to horticultural fields and providing insight into understanding characterization of biosynthetic enzymes in planta. For this reason, I regret your poor manuscript preparation. For example, the plant “Phalaenopsis” with scientific name in the title is not mentioned in the main text. Is this plant identical to “Phalaenopsis” in the main text? References section does not follow Instructions for Author; several of citaitons reffered to in the main text in not listed in References. In general, the term “anthocyanin” is used as singular noun. Nevertheless, authors applied differrent usage in the second paragraph of Introduction section. In Material and Methods section, a description (lines 149-156) is identical to the one (lines 166-173). Several sentenses are written in imperative form, which appears copy and paste of laboratory’s manual. In Results section, the first subheading part includes construction of plasmids and transgenic plants which should be described separately. Primers mentioned in Figure 2 caption are not described in Material and Methods section. There are other points to point out, but they are ommited.

Author Response

Dear editor and reviewers:

On behalf of my co-authors, we thank you very much for giving us an opportunity to revise our manuscript; we appreciate editors and reviewers very much for their positive and constructive comments and suggestions on our manuscript (plants-2304838). We have studied reviewer’s comments carefully and have made revision which marked in red in the paper. We have tried our best to revise our manuscript according to the comments. Attached please find the revised version, which we would like to submit for your kind consideration. We would like to express our great appreciation to you and reviewers for comments on our paper.

Looking forward to hearing from you.

Special thanks to you for your good comments!

Yours sincerely,

Feng Ming

The main corrections in the paper and the responds to the reviewer’s comments are as flowing:

Responds to the reviewer’s comments:

Reviewer #3:

Comments and Suggestions for Authors

The manuscript describes construction of transgenic plants transformed with flavonoid bioisynthetic genes. This approach will contribute to adding breeding options to horticultural fields and providing insight into understanding characterization of biosynthetic enzymes in planta. For this reason, I regret your poor manuscript preparation. For example, the plant “Phalaenopsis” with scientific name in the title is not mentioned in the main text. Is this plant identical to “Phalaenopsis” in the main text?

Response: Thanks for the suggestion. We are sorry that our manuscript was not well prepared. The Phalaenopsis orchids we mention in the body are all Phalaenopsis aphrodite varieties mentioned in the title

References section does not follow Instructions for Author; several of citaitons reffered to in the main text in not listed in References.

Response: The references section has been carefully checked and revised now,We are sorry for these mistakes.

In general, the term “anthocyanin” is used as singular noun. Nevertheless, authors applied differrent usage in the second paragraph of Introduction section. In Material and Methods section, a description (lines 149-156) is identical to the one (lines 166-173).

Response: Thanks for these kind suggestions. The usage of the term anthocyanins has also been corrected; You mentioned that the duplication represents two different tissue culture methods of plant material, and we have carefully considered your suggestion that it needs to be preserved to avoid misunderstanding among readers.

Several sentences are written in imperative form, which appears copy and paste of laboratory’s manual. In Results section, the first subheading part includes construction of plasmids and transgenic plants which should be described separately.

Response: Thanks for these kind suggestions. We checked thoroughly the whole method and changed unclear and repeat manual. Sorry for these faults made. The plasmid construction method was described separately according to your good suggestion and they were marked in red color in method and material section.

Primers mentioned in Figure 2 caption are not described in Material and Methods section. There are other points to point out, but they are ommited.

Response: Thanks for your suggestions. We provide primers of this study in method section. In addition all mediums gradients for protocorm-likebodies(PLB)induction and differentiation and Agrobacterium infection were supplied in Table S1.

Round 2

Reviewer 1 Report

In this manuscript, Phalaenopsis genes related to flower color (PhCHS5 and PhF3'5'H) were transformed into petunias, and the phenotype and anthocyanin content of the resulting transgenic plants were analyzed. The experimental design and manuscript writing are well done, and it is recommended that it can be published in the Plants journal without further modification.

Author Response

Thank you for the affirmation of the article!

Reviewer 2 Report

1) “3) Authors should improve used legends for figures, e.g. explain all used abbreviations, explain all investigated parameters, used statistical treatment

Response: Sorry for missing explanation for abbreviations and short of statistical treatment analysis for figure 5b. We checked the whole article and added explanation. There are no similar mistakes.”

- Answer: Fig. 2, Fig. 5 – I did not find information about used statistical treatment – e.g. ANOVA or Student t-test.

2) “8) Author should improve all presented PCR data, e.g. include figure for actin PCRproducts. Also, I did not understand Fig. 2 g and h.

Response: Thanks for your suggestion. We have replaced the electrophoresis plot in Figure 2 and Figure 5 B and D with a higher definition image. Fig 2g and h represented PhCHS5-transformed and PhF3’5’H-transformed petunia plants were identified by Kan genes PCR detection.”

- Answer: – I did not find PCR products for actin gene.

3) Authors should improve the manuscript title –

Effects of flower color genes PhCHS5 and PhF3'5'H on Petunia (Petunia hybrida) and Phalaenopsis (Phalaenopsis aphrodite)” – describe in title types of effects.

Author Response

1. “3) Authors should improve used legends for figures, e.g. explain all used abbreviations, explain all investigated parameters, used statistical treatment

Response: Sorry for missing explanation for abbreviations and short of statistical treatment analysis for figure 5b. We checked the whole article and added explanation. There are no similar mistakes.”

- Answer: Fig. 2, Fig. 5 – I did not find information about used statistical treatment – e.g. ANOVA or Student t-test.

Response: Sorry for missing explanation for abbreviations and short of statistical treatment analysis for figure 2 and figure 5b. We checked the entire article and added statistical analysis in the Materials & Methods section. There are no similar mistakes.” The added method section is as follows:

Statistical analysis

The date including anthocyanin content, rate of protocorm and branch number was used t-test analysis and the Dumcans multiplerenge test method.

2. “8) Author should improve all presented PCR data, e.g. include figure for actin PCR products. Also, I did not understand Fig. 2 g and h.

Response: Thanks for your suggestion. We have replaced the electrophoresis plot in Figure 2 and Figure 5 B and D with a higher definition image. Fig 2g and h represented PhCHS5-transformed and PhF3’5’H-transformed petunia plants were identified by PCR detection for KAN genes”

- Answer: – I did not find PCR products for actin gene.

Response: Thanks for the suggestion. We tested the KAN gene on the vector to verify that the gene was integrated into the plant's genome, so actin gene was not used.

3. Authors should improve the manuscript title –

“Effects of flower color genes PhCHS5 and PhF3'5'H on Petunia (Petunia hybrida) and Phalaenopsis (Phalaenopsis aphrodite)” – describe in title types of effects.

Response: Thanks for the suggestion. We have changed the title of the article to " PhCHS5 and PhF3'5'H genes over-expression in Petunia(Petunia hybrida) and Phalaenopsis (Phalaenopsis aphrodite) regulate flower color and branch number”.

Reviewer 3 Report

Several points pointed out in the first peer review has not been improved. Some of these are: References section does not follow Instructions for Author. Several citaitons reffered to in the main text do not listed in References: for example, Maarten et al. 2016 (line 36), Katsumoto et al. 2004 (line 60), and so on.

Author Response

Several points pointed out in the first peer review has not been improved. Some of these are: References section does not follow Instructions for Author. Several citaitons reffered to in the main text do not listed in References: for example, Maarten et al. 2016 (line 36), Katsumoto et al. 2004 (line 60), and so on.

Response: Thanks for the suggestion. We have checked and supplemented this reference in the references section. Sorry for these mistakes.

Round 3

Reviewer 3 Report

Several points pointed out in the first peer review has not been improved. One of these is: References section does not follow Instructions for Author. 

Below, I will point out the insufficient description only for construction of expression vectors. There are other points to point out, but they are ommited. For experiments shown in Fig. 1a and 1b, detail procedures are not described in Material and Methods section: What is Cav35s-primerF, CHS-XR’ and CHS-XF primers in Fig. 1 legend?; What were PhCHS5 and PhF3’5’Hgenes amplified using as a template?; What kind of nucleic acid was extracted?; Which stage of the plant was the RNA extracted from, if RNA extraction was performed?; Under what conditions were the reverse transcription and PCR performed?; The reverse primer for PhCHS5 do not have XbaI site, and The forward primer for PhF3’5’Hdo not have BglII site.; I do not understand why both genes and pCAMBIA2301 were digested with XhoI.